# Serum Vitamin D Levels Explored in the Latvian Cohort of Patients with Basal Cell Carcinoma Linked to the Sonic Hedgehog and Vitamin D Binding Protein Cutaneous Tissue Indices

**DOI:** 10.3390/nu14163359

**Published:** 2022-08-16

**Authors:** Jeļena Moisejenko-Goluboviča, Valērija Groma, Šimons Svirskis, Anna Ivanova

**Affiliations:** 1Department of Doctoral Studies, Riga Stradins University, Dzirciema Street 16, LV-1007 Riga, Latvia; 2Institute of Anatomy and Anthropology, Riga Stradins University, 9 Kronvalda Blvd., LV-1010 Riga, Latvia; 3Institute of Microbiology and Virology, Riga Stradins University, Rātsupītes Str. 5, LV-1067 Riga, Latvia; 4Department of Maxillofacial Surgery, Institute of Stomatology, Riga Stradins University, Dzirciema Street 20, LV-1007 Riga, Latvia

**Keywords:** basal cell carcinoma, ultraviolet radiation, vitamin D deficiency, serum levels of vitamin D, Sonic Hedgehog, vitamin D binding protein, immunohistochemistry, hierarchical clustering

## Abstract

Ultraviolet radiation is known as one of the major contributors to skin malignancies, including basal cell carcinoma (BCC), which is the most common type of skin cancer. It is a heterogeneous tumor, which presents with various types that are stratified into low- and high-risk tumors. Sunlight is important for overall health and vitamin D synthesis in the skin, whereas deviations from the optimal level of vitamin D are shown to be associated with the risk of the development of BCC. The accumulating evidence suggests the ability of vitamin D to antagonize the Sonic Hedgehog (SHH) signaling, the key tumor pathway, and play a protective role in the development of BCC. Additionally, a vitamin D binding protein (DBP) is shown to be implicated in the complex regulation of vitamin D. Here, we aimed to explore serum vitamin D in patients with different primary and recurrent BCC of the head and neck and investigate cutaneous DBP and SHH indices, confirmed immunohistochemically in these subjects. According to the results, 94.9% of the Latvian cohort of BCC patients were found to be deficient in vitamin D. No significant differences in serum vitamin D levels were found between genders, primary and recurrent tumors, and different types of BCC. Serum vitamin D was inversely associated with tumor size. Susceptible male individuals with low blood vitamin D levels were recognized at risk of developing aggressive and recurrent BCC confirmed by the use of hierarchical clustering analysis. In smaller tumors with a favorable course, such as superficial and nodular BCC, the association between high DBP and low SHH tissue expression was found, providing supportive evidence of the existence of a link between vitamin D, proteins involved in its metabolism, as exemplified by the DBP and SHH signaling pathway. The assumption of a deficiency in the protective effect of vitamin D in patients with high-risk BCCs was proposed in low DBP and high SHH tissue indices. New extensions to existing knowledge and characterization of the BCC signaling pathways and their cross-talk with vitamin D are warranted when searching for a preferential effect of vitamin D on skin cancer.

## 1. Introduction

Basal cell carcinoma (BCC) of the skin is one of the most common types of skin cancer that commonly affects people with fair skin. It rarely metastasizes, however, it may manifest with severe tissue damage that occurs locally. Clinically and morphologically, BCC presents with various types that are stratified into low- and high-risk tumors. The latter primary BCCs may relapse and display a worse overall prognosis. Commonly, BCC develops in the sun-exposed areas of the skin, such as the head and neck area of elderly people, and the causative role of ultraviolet radiation (UVR) in the development of a tumor has been considered [1,2,3,4]. Globally, a higher incidence of BCC in more equatorial than polar latitudes and areas with significant Caucasian populations has been reported [1,5,6]. Apart from the environmental, occupational, and other risk factors, such as the presence of Fitzpatrick I skin type, family history of skin carcinoma, and immunosuppression, the significance of UVR-induced mutations in the development of BCC has been pointed out [1]. The mutations that activate the Hedgehog intercellular signaling pathway genes, including PTCH, Sonic Hedgehog (SHH), and Smoothened, play a significant role in BCC carcinogenesis [1,3,7,8,9]. Due to the geographic factor, people living in the Baltic region, including the Latvian population, are likely to have a low risk of developing BCC related to chronic ultraviolet exposure; however, these issues remain underexplored until now. Similarly, little is known about the extent of the contribution of solar radiation to the development of more and less aggressive BCC.

Being a substantial nutrient, vitamin D plays a pivotal role in human health. The classical role of vitamin D is associated with the regulation of calcium and phosphorus metabolism and, therefore, the growth and remodeling of bone [10]. Nowadays, the concept of vitamin D is extensively reviewed and revised. It has numerous functions implicated in the complex regulation of physiological processes in the human body [10,11,12,13,14]. Under exposure to ultraviolet B (UVB) rays with a wavelength from 290 up to 315 nm, the synthesis of vitamin D_3_ (cholecalciferol) from 7-dehydrocholesterol (7-DHC) occurs in the skin (Figure 1). Due to its unique feature related to the production of vitamin D_3_ in the skin upon activation by the sun’s UVB rays, vitamin D has been referred to as the sunshine vitamin [15]. Simultaneously, vitamin D_2_ (ergocalciferol) can only be obtained from plant foods, such as yeast, bread, mushrooms, and some vegetables. Taking into account the peculiarities of the synthesis of vitamin D in the skin, which requires intense ultraviolet B-radiation, and the deficiency in the consumption of a sufficient amount of animal products containing vitamin D, the problem of vitamin D deficiency has gained global significance [16]. Furthermore, there is growing concern among professionals that sun protection, recommended by dermatologists to reduce morbidity from skin cancers, can lead to abnormally low levels of vitamin D, which, in turn, could have subsequent adverse effects on the body [17].

Vitamin D is reported to have anti-carcinogenic effects, however, accumulating evidence is controversial [18,19]. The contribution of vitamin D to the prevention of skin malignancies, including BCC, squamous cell carcinoma (SCC), and melanoma, is proven by some studies [20,21]. Subjects with high levels of vitamin D are reported to be less likely to develop non-melanoma skin cancer (NMSC) than people with low levels, and the maintenance of 25-OH vitamin D_3_ levels above 25 ng/mL is shown to significantly reduce the rate of recurrence [22,23,24]. In contrast, no beneficial effect in preventing BCC was shown in a randomized clinical trial of supplementation with vitamin D and/or calcium [25]. Furthermore, the results of other studies suggest that higher serum 25-OH vitamin D_3_ levels are associated with an increased risk of the subsequent development of BCC and melanoma, thus pointing out the inconsistency of current knowledge and the absence of widely applicable strategies [26,27].

The effect of vitamin D_3_ is mediated through the interaction with the nuclear vitamin D receptor (VDR) and retinoic acid orphan receptors (ROR)α and RORγ [13,28]. The 1,25(OH)2D-induced activation of VDR transcription, followed by the enhanced differentiation and reduced proliferation of keratinocytes, was proven [29]. Reversely, decreased VDR expression, found in advanced colorectal and other neoplasms, suggests that loss of VDR may contribute to cancer progression [30]. In addition to expressing the VDR, numerous cells, including keratinocytes, express the vitamin D 1α-hydroxylase, thus permitting local synthesis of the active hormonal form of the vitamin [31]. In turn, the specific transporter, vitamin D binding protein (DBP) contributes to inflammatory and immune processes, binding of actin and circulating fatty acids [32]. The liver cells highly express DBP and export it into the blood [33]. The uptake of circulating DBP is mediated by the endocytotic process in several epithelial cell types [34]. Cell bearing receptor proteins megalin and cubulin can internalize the DBP-bound-25(OH)D complex into an endolysosome and further metabolize and/or catabolize 25(OH)D in the cell interior. Intracellularly, binding proteins can specifically target vitamin D either for association with the VDR and activation or degradation [21]. Notably, factors that influence the levels of megalin or intracellular vitamin D binding proteins could, therefore, potentially alter the cell-specific responses to vitamin D, however, the investigations focused on these issues have not been undertaken as far.

Therefore, we aimed to explore serum vitamin D in patients who presented with different, both less and more aggressive types of BCC, and investigate cutaneous DBP and SHH indices confirmed in these subjects.

## 2. Materials and Methods

### 2.1. Patients’ Characteristics and BCC Classification

In total, 79 patients clinically presented with the suspected BCC of the head and neck treated prospectively in Riga Stradins University, Institute of Stomatology, Department of Maxillofacial Surgery, and the Oncology Centre of Latvia between September 2016 and September 2019 were used in this study. The age range of subjects was 37–90 years. Among all BCC patients, 46 were women and 33 were men. Clinical data included information on patients’ characteristics, clinical outcomes, complications of BCC, and findings of dermoscopic imaging. Characteristics of a neoplasm included information on the duration and type of the BCC lesion at the time of presentation, anatomical localization, and the size of the tumor. Patients with vitamin D deficiency were prescribed vitamin D therapy, depending on the severity of its deficiency. The disease relapse was monitored over a 2-year follow-up period.

The study was approved by the Ethical Committee of Riga Stradins University (Decision No. 11/08.09.2016.), and written informed consent was obtained from all patients included in the research. The tumor tissue samples were obtained following the tenets of the Declaration of Helsinki.

In all cases, BCC was confirmed, and different types of a tumor were distinguished according to the international classification. The latest WHO classification of skin tumors states that BCC is divided into two types based on the risk of the development of a complicated course and further treatment recommended: 1. lower risk tumors: nodular, superficial, pigmented, and infundibulocystic (a variant of BCC with accessory differentiation), fibroepithelial; 2. higher risk tumors: basosquamous carcinoma, sclerosing/morphoid, infiltrating, BCC with sarcomatoid differentiation, and micronodular [35].

### 2.2. Assays Used for the Detection of Serum Vitamin D Levels

Blood samples for the assessment of vitamin D levels were prospectively collected from all BCC patients and further transferred to the certified E. Gulbis Laboratory Ltd (LATAK accreditation ISO 15189). A conventional chemiluminescence immunoassay using the Cobas 8000 analyzer (Roche, Basel, Switzerland) was performed to measure a total vitamin D serum level [36,37,38].

### 2.3. Desmoscopic Examination Used to Diagnose BCC and Its Assessment Criteria

The dermoscopic examination was performed with a handheld dermatoscope (3Gen DermLite DL3N with PigmentBoost; OlympusDermLite LLC, San Juan Capistano, CA, USA) using a 30 mm × 10 lens before a tumor mass excision. Both contact and non-contact techniques and a polarized mode were used to visualize BCC lesions. A digital photography of the dermoscopic presentation of the BCC lesion was performed using a Samsung Galaxy S9+ (Samsung Electronics, Seoul, Korea) mobile camera.

The diagnosis of BCC was based on the following dermoscopic criteria: the presence of ulceration, maple-leaf-like structures, blue-gray globules, blue-ovoid nests, arborizing vessels, and spoke-wheel structures. Additionally, we used to include the following criteria: translucency, white areas, and milky pink or red background when confirming the presence of BCC. Vascular patterns of the lesions were described as clustered, diffuse, and homogeneous. Background differences between white-red colors were referred to as white-red structureless areas [17].

### 2.4. Histopathological and Immunohistochemical Methods Used to Assess the Cutaneous Tissue Expression of SHH and DBP

In this study, the surgically excised BCC masses were further processed as the formalin-fixed, paraffin-embedded (FFPE), and conventionally sectioned tissue samples. The sections were mounted on adhesive SuperFrost Plus glasses (Gerhard Menzel GmbH, Brunswick, Germany) to better save the tumor tissues exposed to immunohistochemistry. Initially, the sections were routinely stained with hematoxylin and eosin (HE) to confirm the diagnosis and detect the type of BCC. The histopathology of the tumor was assessed by two independent observers.

For the detection of SHH signaling protein and DBP, the tumor tissue sections were deparaffinized in xylene and hydrated in a series of graded ethanol. An endogenous peroxidase blocking process (5 min) was conducted with 0.3% (*v*/*v*) H_2_O_2_ in 95% methanol. The sections were further placed in 10 mM sodium citrate buffer, pH 6.0, at 96–98 °C for 5 min in a microwave for heat-induced epitope retrieval. SHH- and DBP-positive cells were identified by the incubation overnight (4 °C) with the rabbit anti-human monoclonal SHH (Abcam, Cambridge, MA, USA, clone EP1190Y, 1:200), which recognizes the full length and c-product subunit of human SHH protein, and the rabbit anti-human polyclonal DBP (Bioss Antibodies, MA, USA, 1:300), which detects endogenous DBP proteins, and visualized using the HiDef Detection HRP Polymer system (CellMarque, Rocklin, CA, USA). Thereafter, the sections were rinsed and incubated with the HiDef Detection™ Amplifier and the HiDef Detection™ HRP Polymer Detector for 10 min each, respectively. The SHH and DBP immunostaining protocols were adapted from those published previously [39]. The tumor tissue antigenic sites were visualized with 3, 30 diaminobenzidine (DAB) tetrahydrochloride kit (DAB+Chromogen and DAB+Substrate buffer, Cell Marque, Rocklin, CA, USA) applied for 5 min. The cell nuclei were counterstained with Mayer’s hematoxylin. The sections were washed, dehydrated, cleared, mounted in Roti^®^ Histokitt (Carl Roth, Karlsruhe, Germany), and coverslipped. Cells that were labeled by the anti-SHH and anti-DBP antibody and displayed brown reaction products were considered immunopositive. The substitution of the primary antibody with tris(hydroxymethyl)aminomethane (TRIS) solution was used in negative IHC controls.

The stained slides were scanned with a Glissando Slide Scanner (Objective Imaging Ltd., Cambridge, UK). Bright-field images were generated and analyzed by a Leica light microscope (LEICA, LEITZ DMRB, Wetzlar, Germany) using a DFC 450C digital camera. The assessment of immunostaining was performed semi-quantitatively in 10 randomly selected visual fields of each sample (magnification 400×) representing the regions of interest as described previously [9].

### 2.5. Statistical Data Analysis

Statistical data analysis and plotting were performed using Prism 9 software for the macOS (GraphPad Software, LLC, San Diego, CA, USA) and JMP Pro 16 (SAS, Cary, CANC, USA). A two-sample *t*-test was performed to examine the relation between sex, and different prognostic groups (primary and recurrent, low risk and high risk) of BCC and serum D vitamin level. Spearman rank correlation was applied to detect the possible correlation between serum vitamin D level and tumor size, as well as the relation between sex and type of tumor. Pearson’s correlation coefficient was used to estimate the relationships between the immunostaining patterns. For all statistical tests used, a significance level of *p* < 0.05 was chosen. Multivariate analysis and a hierarchical clustering method were used to determine the main subsets of the results obtained.

## 3. Results

### 3.1. General Information, Contact and Non-Contact Desmoscopic Examination of the BCC Lesions

Collectively, a cohort of 79 patients diagnosed with BCC was used in this study. Among all patients who presented with BCC, 58% were women and 42% were men. The mean age of women was 70 years (SD ± 15) and for men it was 64 years (SD ± 17). Anatomically, BCC tumors often were localized on the nose and nasolabial folds, followed by the localization on the cheeks and other areas of the face and neck.

Dermoscopically, the common criteria characteristic of BCC included pink or milky-pink and milky-red color structureless areas, erosions, ulcerations, short thin telangiectasias, arborizing vessels, blue-gray globules, maple leaf-like structures, white streaks, and translucency. The specific dermoscopic diagnostic criteria for the superficial type of BCC included the presence of short thin teleangiectasias and comma vessels, small ulcerations, pink homogeneous areas, and blue-gray ovoid nests (Figure 2A). Nodular and micronodular types were presented with the border raised above the central part of the lesion, arborizing vessels, ulceration, blue-gray ovoid nests, and translucency (Figure 2B,C). Infiltrative BCC was commonly presented with arborizing vessels, short thin telangiectasia, shiny white structureless areas, ulceration, and white streaks (Figure 2D). Finally, the diagnosis of mixed BCCs was based on histopathological observations and the presence of certain dermatoscopic characteristics that included such standard features of BCC as arborizing vessels, ulcerations, milky-red color structureless areas, and less commonly found signs, such as masses of keratin, superficial scaling, and white streaks (Figure 2D).

### 3.2. Assessment of Serum Vitamin D Levels

The assessment of serum vitamin D levels revealed that women presented with a range of 5.9–40.0 ng/mL, whereas men presented a range of 9–32 ng/mL, respectively (Figure 3A). In women, the mean value of vitamin D level was 17.9 ng/mL (SD ± 6.7 ng/mL), whereas, in men, the mean value was 17.7 ng/mL for men (SD ± 7.0 ng/mL), however, these differences were not statistically significant. Only 4 patients in the study cohort had enough serum vitamin D levels estimated above 30.0 ng/mL. In patients presenting with primary BCCs, statistically significant differences in serum vitamin D levels when assessed in men and women were not found (Figure 3B). Simultaneously, in 2 women diagnosed with recurrent BCC tumors, the mean value of vitamin D level was 13.5 ng/mL (SD ± 5.0 ng/mL), whereas, in 16 men, the mean value was 17.0 ng/mL for men (SD ± 6.7 ng/mL), however, these differences were not statistically significant. Serum vitamin D levels were significantly lower in women presented with mixed BCC tumors when compared to men (*p* < 0.05). Simultaneously, serum vitamin D levels did not differ when patients of both genders presented with different types of BCC were compared (Figure 3C).

For a better understanding of the spectrum of serum vitamin D levels observed in BCCs, the indices were analyzed separately for primary and recurrent, as well as for low-risk (superficial and nodular) and high-risk (micronodular, infiltrative, and mixed) tumors. Statistically significant differences in serum vitamin D levels were not observed when low-risk and high-risk BCCs diagnosed in both genders were compared (Figure 4). Similarly, there was not a statistically significant difference found when the results of the assessment of serum vitamin D levels were compared for patients presented with either different types or primary and recurrent BCC tumors (Figure 5).

As depicted in the dendrogram presented in Figure 6, the data were separated into three main branches using the hierarchical clustering method. The blue branch was characteristically represented by males with low serum vitamin D levels and large, mostly high risk and recurrent BCC tumors. In turn, the green branch was represented by both males and females with greatly varying serum vitamin D levels and small-sized primary, however, high-risk BCC tumors. Finally, the red branch was represented by males and females with varying serum vitamin D levels and small-sized primary and less aggressive BCC tumors. Furthermore, in general, a negative association between the tumor size and serum vitamin D level was demonstrated by the use of correlation analysis (Figure 7).

### 3.3. Histopathology, SHH, and DBP Immunohistochemistry of the BCC Tissue Samples

Histopathologically, BCC cells formed nests, cords, and islands. The tumor cells displayed little pleomorphism, and mitotic figures were infrequent. The presence of peripheral palisading, artefactual clefting, and myxoid stroma was characteristic of the BCC tumor. In superficial BCC, a superficial proliferation of neoplastic cells was demonstrated. A nodular type presented with the nodules displaying the typical nuclear palisading at the periphery of the tumor, whereas a micronodular type demonstrated a cauliflower appearance. In infiltrative BCC, small aggregations or cords of basaloid cells were found penetrating the stroma, interspersing muscular fibers, and revealing perineural invasion. The infiltrative growth often showed heavy stromal collagenization. Mixed growth BCC commonly revealed an admixture of rounded and irregularly contoured tumor cell nests and cords embedded in a fibrous stroma. In addition, the presence of mitotic figures and apoptotic cellular debris was characteristic of the aggressive growth.

A pattern of DBP and SHH immunostaining varied when different types of BCC were compared. To better assess the possible associations between the expressions of cutaneous tissue markers, a correlation analysis was performed separately for different BCC types. Assessments of cutaneous DBP and SHH indices observed in low and high risk BCCs by the use of IHC assays and correlation analysis are depicted and summarized in Figure 8 and Figure 9. In superficial BCC, mostly moderate expression of both tissue markers was revealed in superficial BCC (Figure 8A,B). Furthermore, in superficial BCC, a strong negative correlation between SHH and DBP expression in both tumoral *(r* = −0.77) and stromal (*r* = −0.49) compartments was found (Figure 8C). It is worth noting that, weak DBP expression was commonly demonstrated in aggregations of basaloid cells and peripheral cells of the tumor nodules with scanty cytoplasm that displayed the typical nuclear palisading. In contrast, moderate to strong expression of DBP was found in differentiated tumor cells (Figure 8D). Simultaneously, a weak expression of SHH was demonstrated in nodular BCC masses (Figure 8E). The only natural intrinsic correlation was found for the DBP marker in nodular BCC (Figure 8F).

Overall, a more pronounced expression of SHH was observed in tumor cells with low expression of DBP, and vice versa (Figure 9A,B,D,E,G,H). In both infiltrative and mixed BCCs, high SHH expression was found in the tumor tongues presented as admixtures of rounded nodules and nodules with irregular contours, and small irregular tongues of tumor cells embedded in the fibrous stroma (Figure 9B,H). Characteristically, high risk infiltrative BCCs, often with scanty cytoplasm, poorly expressed DBP (Figure 9G). Intrinsic correlations were found for the DBP and SHH markers presented in the tumoral and stromal compartment of a neoplasm when assessing high risk BCCs by the use of statistical data analysis (Figure 9C,F). Simultaneously, in infiltrative BCC, a negative correlation between DBP and SHH stromal expression (*r* = −0.56) was found (Figure 9I).

Finally, to better assess the complex associations between serum vitamin D level patterns detected in the Latvian cohort of patients presenting with the different types of BCC and the cutaneous tissue expression of DBP and SHH confirmed by the use of IHC in these individuals, a Spearman’s rank correlation analysis was performed (Figure 10A–C). Similarly, to the results obtained and interpreted when describing a dendrogram (Figure 6), in this final assessment, males recruited in the given study characteristically presented with lower than females serum vitamin D levels, high risk (*r* = −0.87), and recurrent (*r* = −0.87) BCC tumors.

## 4. Discussion

In the present study, the serum levels of vitamin D were explored in the Latvian cohort of patients who presented with different primary and recurrent BCC of the head and neck; cutaneous DBP and SHH indices found in these subjects were investigated, and established correlations were analyzed. According to the results of the present research, 94.9% of the Latvian cohort of patients with primary and recurrent BCC of the head and neck were found to be deficient in vitamin D. In total, 5.1% of the examined patients had a sufficient level of 30.0–40.0 ng/mL. These data are in agreement with the results of the vitamin D deficiency study conducted in Latvia in 2015 by specialists from the Latvian Osteoporosis and Bone Metabolic Diseases Association, Riga Stradins University, and Riga East Clinical Hospital which is the only study that has been performed before in Latvia [40]. The results of the aforementioned study and the drawn conclusions have highlighted the presence of vitamin D deficiency in 82% of the study participants. Similar results come from the neighboring countries reporting vitamin D deficiency when carrying out a large Swedish cohort study and confirming the average estimated level of vitamin D as 19.9 ng/mL [41], thus being slightly higher than that determined in the Latvian cohort of BCC patients. In turn, the Lithuanian colleagues confirmed the presence of seasonal vitamin D deficiency in 67% of outpatient department subjects investigated retrospectively [42]. Furthermore, according to the results of the study undertaken by the National Institutes of Health using the developed protocols to standardize the existing 25(OH)D values from National Health/Nutrition Surveys in a frame of the International Vitamin D Standardization Program, the prevalence of vitamin D deficiency assessed at 55,844 Europeans is 40.4% [43]. These data are also supported by another large cohort study suggesting vitamin D deficiency occurs in <20% of the populations in Northern Europe, in 30–60% in Western, Southern, and Eastern Europe, and up to 80%—in the Middle Eastern countries, severe deficiency (<12  ng/mL) is found in >10% of Europeans [44]. Based on the results of the analysis of pertinent literature and present research, it is safe to say that in Europe, in general, and in the Baltic region, in particular, the serum vitamin D level can be described as catastrophically low. The knowledge accumulated to date about the role of vitamin D in the human body and the relationship of its level with the development and course of oncological pathology suggests that the correction of insufficient levels of vitamin D may be relevant in the prevention and complex treatment of cancer. The recommendation by the Institute of Medicine for serum 25(OH)D targets of 50–75 nmol/L appears prudent and is supported by skin cancer association studies [45]. Vitamin D supplementation is shown to be beneficial in the reduction in cancer, 25(OH)D concentrations ≥ 60 ng/mL markedly lower the risk of cancer [46,47]. The results of this study highlight the necessity of educative measures related to the effects of sunshine on the skin and raise awareness about vitamin D deficiency in the general population.

Deficient levels of 25(OH)D have been found in tumors with the locally destructive and metastatic course, tumors with the involvement of lymph nodes, and neoplasms demonstrating higher proliferative activity [48]. Low levels of vitamin D determined in menopausal patients were consistent with the presence of advanced neoplastic disease, larger tumor size, and its grade [20]. A significant association between metastatic breast cancer and vitamin D levels is shown [49,50]. Low serum DBP levels are thought to predict lung cancer-specific death, and preservation of serum DBP was recognized as a significant independent factor associated with better cancer outcomes in operated lung cancer patients [51]. Of note, deviations from the optimal level of vitamin D have been associated with the risk of the development of BCC [19,52].

In the present study, conducted using a medium-sized cohort of BCC patients, a cluster tree stratified into three major branches was recognized. The first branch was represented by males with low serum vitamin D levels and large, mostly high risk and recurrent BCC tumors. Therefore, it can be said with certainty that male sex and low blood vitamin D levels are risk factors for the development of aggressive types of BCC. The second branch was represented by both males and females with greatly varying serum vitamin D levels and small-sized primary, however, high risk BCC tumors. Finally, the third branch was represented by males and females with varying serum vitamin D levels and small-sized primary and less aggressive BCC tumors.

Among several signaling pathways which exert their effects on BCC pathogenesis, the hedgehog pathway has been intensively investigated. The body of evidence suggests that vitamin D inhibits the SHH signaling pathway and, therefore, displays a protective role in arresting tumor growth [52,53,54,55]. D_3_ specifically binds to Smo and thereby inhibits the activity of Gli proteins in fibroblasts in vitro, suggesting that the Ptch1 protein performs its suppression of SHH signaling by transporting vitamin D_3_ to the Smo protein [56,57,58,59]. Vitamin D can affect skin cancer cells also through interaction with the immune system cells. Calcitriol inhibits type 1 (Th1) helper T cells and increases Th2 and regulatory T cell responses, which reduces immune surveillance for skin cancer [20,56]. In turn, there is an increase in innate immunity and the induction of Toll-like receptors, which possibly improves anticancer responses [57]. Pertinent literature suggests that the intracrine metabolism of 25(OH)D to 1,25(OH)_2_D is likely to occur in several tissue cells, including keratinocytes, that express VDR and harbor a specific enzyme CYP27B1 [32]. Furthermore, some epithelial cells, including mammary gland and renal tubular cells express megalin and cubilin, which contribute to the endocytic uptake of 25(OH)D_3_-DBP and activation of the VDR pathway [60,61]. Intracellularly, 25(OH)D_3_-DBP binds the cytosolic actin to create a complex.

Vitamin D and its metabolites are bound to a specific transporter—DBP, which belongs to the albuminoid family of proteins and ensures the storage/transport of vitamin D and fatty acids, scavenging of extracellular G-actin, enhancement of the chemotactic activity of C5 alpha for neutrophils, and contributes to other physiological processes [34,62]. Although serum DBP is a protein that displays multiple properties, including the regulation of vitamin D levels, its functional implications largely remain unknown. In turn, the tissue expression of DBP is likely to be associated with favorable prognostic characteristics, such as small tumor size, and low invasiveness. This is consistent with the evidence that DBP-positive tumors are associated with a reduced risk of metastasis and mortality from various cancers. Recent studies that are in line with the results of the present research demonstrate a significantly low to no-DBP immunohistochemical staining in advanced tumors [63]. In contrast, moderate to strong immunostaining was found to correlate with early-stage tumors. The possible stimulation of an intracellular immune-modulating signaling pathway in thyroid cancer oncogenesis in the loss-of-DBP-function in the tumor tissues was suggested [63]. In the present study, statistically significant differences in vitamin D levels between males and females, and different types of BCC were not found. Simultaneously, a correlation between the tumor size and serum vitamin D levels was established, and this observation is in agreement with data from other studies [64]. Furthermore, the association between high DBP and low SHH tissue expression found in smaller tumors with a favorable course, such as superficial and nodular BCC, was demonstrated. Importantly, low expression of DBP and high expression of SHH were observed in mixed and infiltrative BCCs which may correlate with the assumption of a deficiency in the protective effect of vitamin D in patients with high-risk tumors and a tendency to relapse after treatment. Simultaneously, it becomes obvious that the link between exposure to sunlight and BCC is not straightforward, as high levels of intermittent UV exposure appear to be more associated with the development of a tumor in susceptible individuals rather than long-term exposure, as seen in long-term outdoor workers [65,66].

The results of the given study have to be viewed in light of some limitations. A moderate sample size of the study cohort may be assumed as a potential limitation. The disease relapse was monitored over a 2-year follow-up period only. Simultaneously, the relevance of the results that are obtained and conclusions that are drawn out is enhanced by the prospective nature of the present research. Only a conventional chemiluminescence immunoassay was used to measure the total vitamin D serum level in BCC patients. Finally, the size of BCC tissue obtained by surgical excision in the head and neck region was strongly associated with the treatment necessity, and, therefore, explains some difficulties related to the number of histopathology and immunohistochemistry assays performed.

## 5. Conclusions

Vitamin D deficiency was determined in the Latvian cohort (94.9%) of patients with both primary and recurrent BCC of the head and neck. Higher serum vitamin D levels correlate with the appearance of smaller neoplasms and a more favorable disease prognosis. In smaller tumors with a favorable course, such as superficial and nodular BCC, the association between high-DBP and low-SHH tissue expression was found, providing supportive evidence of the existence of a link between vitamin D, proteins involved in its metabolism, as exemplified by the DBP, and SHH signaling pathway. The assumption of a deficiency in the protective effect of vitamin D in patients with high-risk BCCs was proposed in low-DBP and high-SHH tissue indices. New extensions to existing knowledge and characterization of the BCC signaling pathways and their cross-talk with vitamin D are warranted when searching for a preferential effect of vitamin D on skin cancer.

## Figures and Tables

**Figure 1 nutrients-14-03359-f001:**
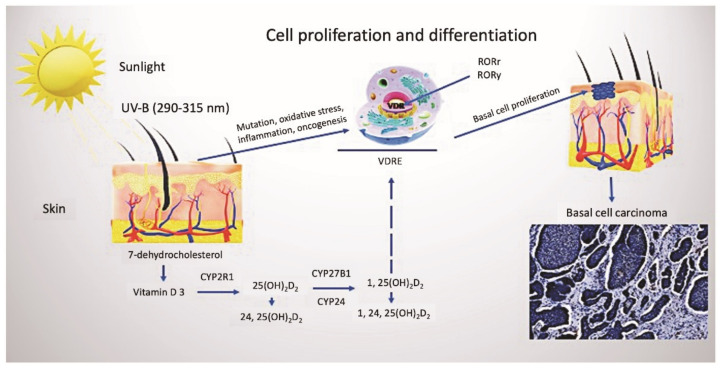
Vitamin D metabolism and cutaneous synthesis. Under exposure to UVB rays, the process of vitamin D_3_ synthesis proceeds in keratinocytes, which contain CYP enzymes necessary for the photochemical conversion of the precursor of vitamin D_3_, 7-dehydrocholesterol, into its active form (calcitriol 1, 25(OH)_2_D_2_). The activities of hydroxyderivatives of vitamin D are mediated by the involvement of the ligand-binding domain of the nuclear receptor, VDR. Furthermore, vitamin D_3_ is implicated in the regulation of biological functions and gene expression of keratinocytes of both healthy and BCC-affected subjects, mediated by the presence of ROR α and γ nuclear receptors. Abbreviations: UVB, ultraviolet B; VDR, vitamin D receptor; RORα and γ, retinoic acid-related orphan receptors α and γ; BCC, basal cell carcinoma.

**Figure 2 nutrients-14-03359-f002:**
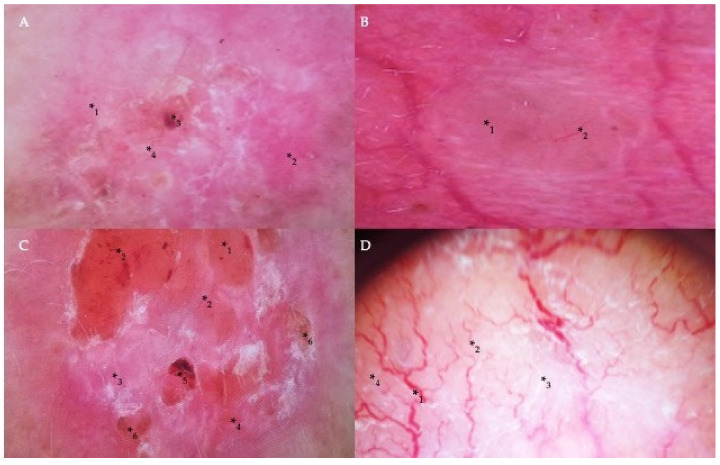
Dermoscopic findings in different types of BCC. (**A**) Milky-pink color structureless areas (*_1_), pink homogeneous areas (*_2_), short thin telangiectasia (*_3_), and erosion (*_4_) on the central part of the tumor in the case of superficial BCC. (**B**) A translucent nodule (*_1_) raised above the skin and arborizing vessels (*_2_) in nodular BCC. (**C**) The nodular appearance of a tumor with a border raised above the central part of the lesion (*_1_), arborizing and short thin telangiectasias (*_2_), milky-pink (*_3_) and milky-red color structureless areas (*_4_), ulceration (*_5_), and erosion (*_6_) in the case of micronodular BCC. (**D**) Arborizing vessels (*_1_), short thin telangiectasia (*_2_) on shiny white (*_3_) and milky-red structureless areas (*_4_) in the infiltrative and mixed type of BCC.

**Figure 3 nutrients-14-03359-f003:**
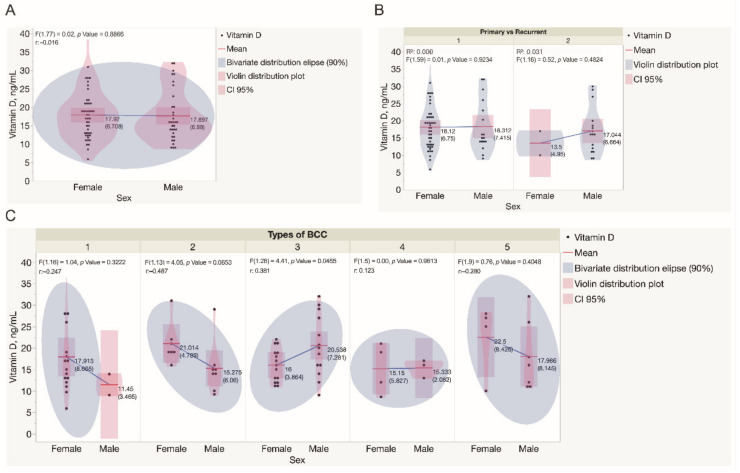
Assessment of serum vitamin D levels in the study cohort. (**A**) Assessment of serum vitamin D levels in males and females recruited in the study. Each dot represents a single data point. (**B**) Assessment of serum vitamin D levels in males and females presented with either primary or recurrent BCC tumors. Each dot represents a single data point. (**C**) Assessment of serum vitamin D levels in males and females presented with different types of BCC tumors. Each dot represents a single data point. Statistically significant differences in serum vitamin D levels were not observed when different types of BCC diagnosed in both genders were compared, except in mixed BCC tumors, diagnosed in females presented with significantly lower serum vitamin D levels when compared to males (*p* = 0.0456). Abbreviations: (**B**) 1—primary BCC tumors; 2—recurrent BCC tumors; (**C**) 1—superficial BCC tumors; 2—nodular BCC tumors; 3—mixed BCC tumors; 4—micronodular BCC tumors; 5—infiltrative BCC tumors.

**Figure 4 nutrients-14-03359-f004:**
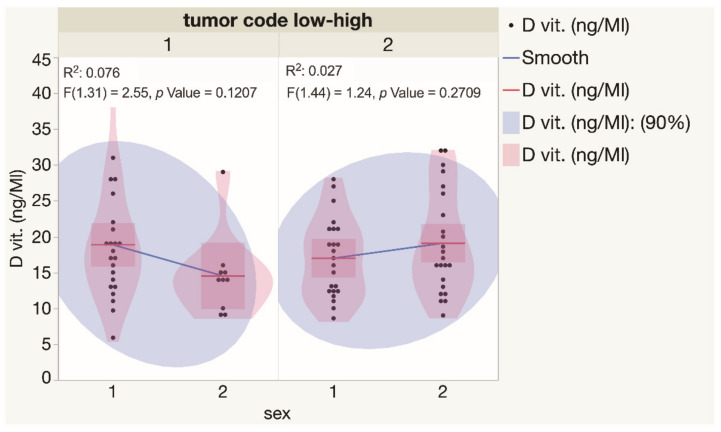
Assessment of serum vitamin D levels in males and females presented with low and high risk BCCs. Each dot represents a single data point. Abbreviations: 1—low-risk BCC tumors; 2—high-risk BCC tumors.

**Figure 5 nutrients-14-03359-f005:**
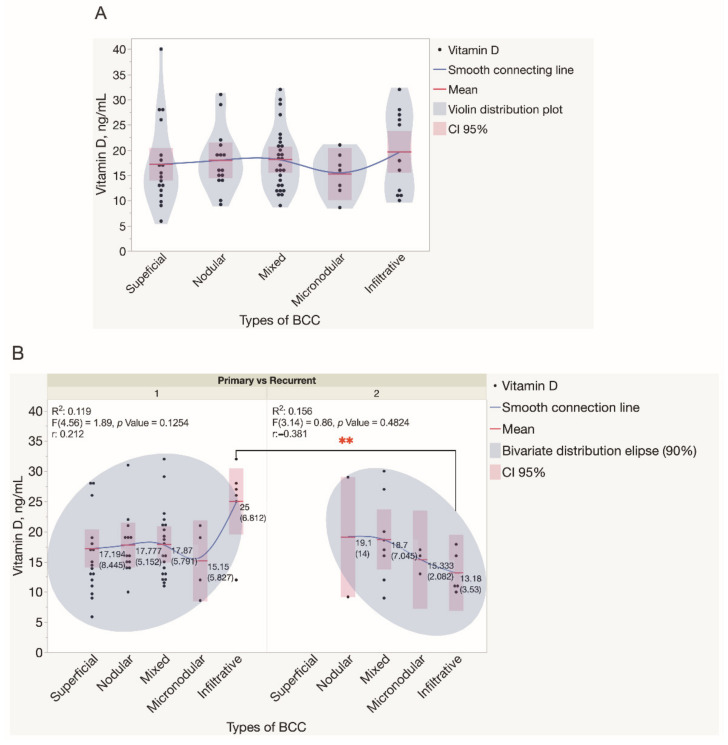
Assessment of serum vitamin D levels in patients presented with different types of BCC tumors. (**A**) Violin distribution plots depict the results of the assessment of serum vitamin D levels in patients presented with different types of BCC tumors. Each dot represents a single data point. (**B**) The results of the assessment of serum vitamin D levels in patients presented with either primary or recurrent BCC tumors. Each dot represents a single data point. Abbreviations: (**B**) 1—primary BCC tumors; 2—recurrent BCC tumors. ** *p* < 0.01.

**Figure 6 nutrients-14-03359-f006:**
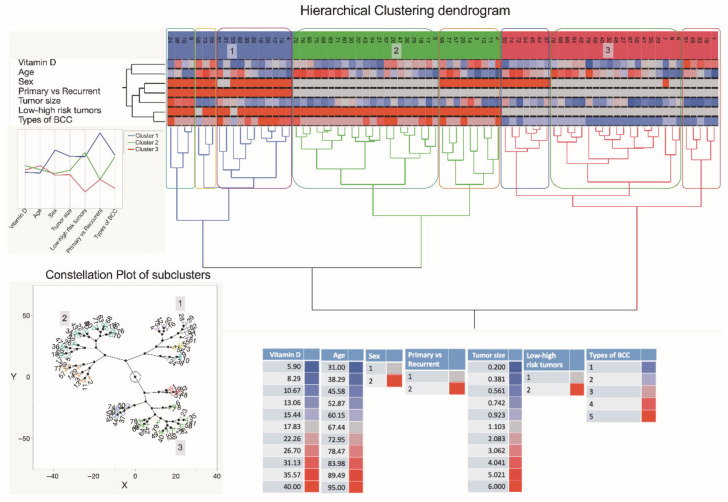
A dendrogram shows hierarchical clustering—relationships between sets of data. The dendrogram consists of stacked branches (clades) that break down into further smaller branches. At the lowest level, individual elements appear and then they are grouped according to attributes into clusters with fewer and fewer clusters on higher levels. The end of each clade (a leaf) are the data. The sets of data included assessments of serum vitamin D level patterns of 79 subjects presenting with the different types of BCC, information about a patient age and sex, a tumor type (primary vs. recurrent), and a type of BCC, and seen in the right lower part of the Figure. The tumor specimens were divided into five subtypes based on differences in histopathology and dermoscopic imaging. The three major data clusters with 2–3 subclusters each are highlighted in the constellation plot of the dendrogram depicted in the left lower part of the Figure. Abbreviations: sex: 1—females; 2—males; primary vs. recurrent BCC: 1—primary BCC; 2—recurrent BCC; low and high-risk BCC: 1—low risk BCC; 2—high risk BCC; types of BCC: 1—superficial BCC tumors; 2—nodular BCC tumors; 3—mixed BCC tumors; 4—micronodular BCC tumors; 5—infiltrative BCC tumors.

**Figure 7 nutrients-14-03359-f007:**
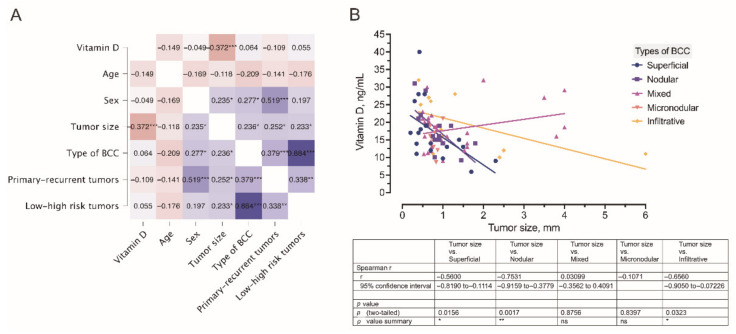
(**A**) A correlogram of the studied variables. In this plot, correlation coefficients are colored according to the value. Positive correlations are displayed in blue, whereas negative correlations are in red. Color intensity is proportional to the correlation coefficients. The negative association (r = −0.372) between the tumor size and serum vitamin D level is marked by a dark red color. (**B**) Correlation between the tumor size and serum vitamin D level was observed in different types of BCC. * *p* < 0.05; ** *p* < 0.01; *** *p* < 0.001.

**Figure 8 nutrients-14-03359-f008:**
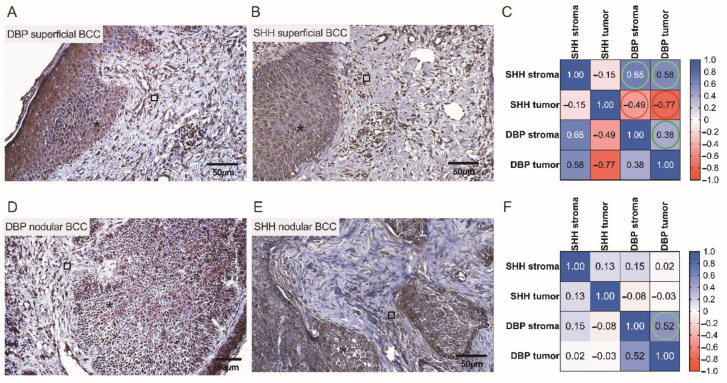
Assessment of cutaneous DBP and SHH expression in low risk BCCs by the use of IHC. Representative images demonstrating cutaneous structures (tumoral (⁎) and stromal (□)) decorated by the anti-DBP (**A**,**D**) and anti-SHH (**B**,**E**) antibodies and recognized by the presence of brown reaction products found in the BCC samples. Scale bars: 50 μm. Correlograms that highlight associations between the expression of DBP and SHH in superficial (**C**) and nodular (**F)** BCCs. A strong negative correlation between SHH and DBP expressions is marked by red color (**C**).

**Figure 9 nutrients-14-03359-f009:**
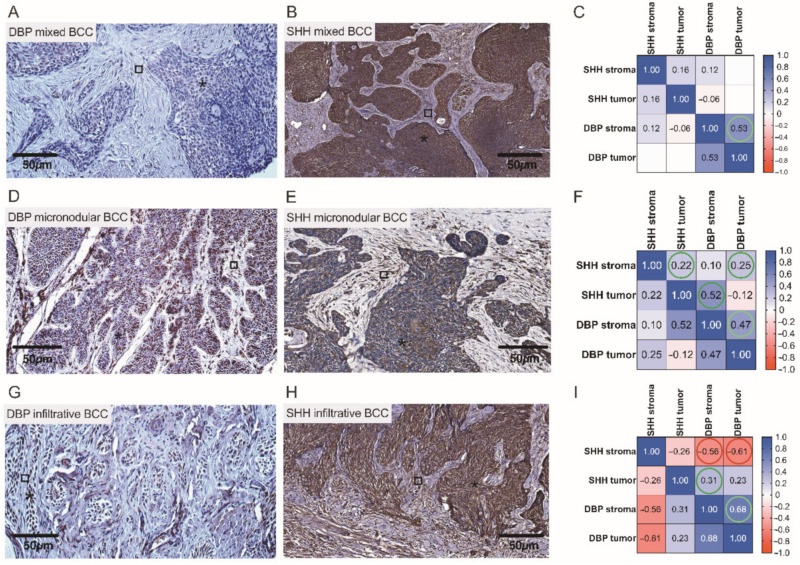
Assessment of cutaneous DBP and SHH expression in high risk BCCs by the use of IHC. Representative images demonstrating cutaneous structures (tumoral (⁎) and stromal (□)) decorated by the anti-DBP (**A**,**D**,**G**) and anti-SHH (**B**,**E**,**H**) antibodies and recognized by the presence of brown reaction products found in the BCC samples. Scale bars: 50 μm. Correlograms that highlight associations between the expression of DBP and SHH in mixed (**C**), micronodular (**F**), and infiltrative (**I**) BCCs. A negative correlation between SHH and DBP expressions is marked by red color (**I**).

**Figure 10 nutrients-14-03359-f010:**
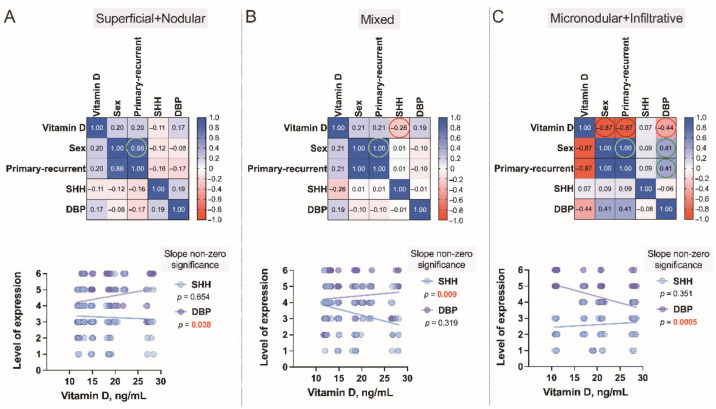
Correlograms of the studied variables, Superficial+Nodular (**A**), Mixed (**B**) and Micronodular+Infiltrative (**C**). The correlograms depict the complex associations between serum vitamin D level patterns of 79 subjects presenting with the different types of BCC and the cutaneous tissue expression of vitamin D binding protein (DBP) and Sonic Hedgehog (SHH) confirmed by the use of IHC in these individuals. Each dot represents a single data point. The levels of expression are assessed as a sum of the tumoral and stromal BCC tissue indices. In these plots, correlation coefficients are colored according to the value. Positive correlations are displayed in blue, whereas negative correlations are in red. Color intensity is proportional to the correlation coefficients.

## Data Availability

A publicly available bibliographic database, PubMed.gov, was used in this study. The complete search query is specified in the Methods section of the article. The full bibliographic reference list is available on request from the corresponding author.

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
