# Peer review of "Serum Vitamin D Levels Explored in the Latvian Cohort of Patients with Basal Cell Carcinoma Linked to the Sonic Hedgehog and Vitamin D Binding Protein Cutaneous Tissue Indices"

_nutrients, 2022, doi:10.3390/nu14163359_

Round 1

Reviewer 1 Report

The manuscript by Jeļena Moisejenko-Goluboviča et al. examined the serum levels of vitamin D in patients with primary and recurrent BCC of the head and neck and link these data to the results of tissue SHH and DBP immunohistochemistry assessment. They found that there are no significant differences in serum vitamin D levels were found between genders, primary and recurrent tumors, and different types of BCC.

The major results and conclusions of this research are not clear and novelty. The study requires improvement in numerous aspects. Please consider the following points:

1. Most of the Abstract introduced the background of the study in line 15-23, the results was showed too little. The conclusion was not clear. What is the meaning via the detection of serum vitamin D.

2. The introduction was too long. I did not understand your aim of this study and see the scientific question you want to solve. According to your abstract, first you should introduce the background of basal cell carcinoma (BCC), second the relationship between BCC and vitamin D, third the role of vitamin D, forth the effects and mechanisms of action of vitamin D on BCC, finally raising your scientific questions and potential meaning.

3. The methods are too single. The levels of SHH and DBP should also be detected by Western blot, real-time quantitative PCR, and immunofluorescent staining.

4. The results were not described clearly, especially the figure. In figure 2, the common criteria characteristic of BCC was not indicated in the figure 2A-D by arrows or others. In figure 6 and 7, the date was not described in the figures. Similarly, the same problem accrued in figure 8-10.

5. The Discussion was mainly introduced previous study, did not see your discussion combined with your results.  According to your discussion, low level of serum vitamin D had been demonstrated in Latvia in 2015 by specialists from the Latvian Osteoporosis and Bone Metabolic Diseases Association, Riga Stradins University, and Riga East Clinical Hospital [38]. So we did not see your novelty, you should discuss your new findings in the discussion. Similarly, you detected the levels of SHH and DBP. As you known, vitamin D inhibits the SHH signaling pathway and displays a protective role in arresting tumor growth [37].  According to the previous report, the level of SHH increased when vitamin D decreased. Where is your new findings.   

6. There are some spelling mistakes, such as H2O2 in line 196 and 4 ◦C in line 199.

7. The description of results should be in sequence, figure 9 should be ahead figure 8, because you described figure 9 first in line 346. Similarly, in figure 8 you described figure 8D first in line 350.

Author Response

The authors thank the reviewer for his noticeable expertise in this field and a thoughtful and improvement-driven review of the manuscript. To the best of our knowledge and possibilities, we have improved the manuscript as suggested.

The 1st comment: Most of the Abstract introduced the background of the study in line 15-23, the results was showed too little. The conclusion was not clear. What is the meaning via the detection of serum vitamin D.

Response: We thank the Reviewer for these constructive comments. According to the Reviewer’s request, the Abstract and Conclusions were redrafted and rewritten.

The 2nd comment: The introduction was too long. I did not understand your aim of this study and see the scientific question you want to solve. According to your abstract, first you should introduce the background of basal cell carcinoma (BCC), second the relationship between BCC and vitamin D, third the role of vitamin D, forth the effects and mechanisms of action of vitamin D on BCC, finally raising your scientific questions and potential meaning.

Response: We have restructured, rewritten, and shortened the Introduction according to the Reviewer’s request. We have added a more detailed description (supported by some recent publications, including those that appeared in Nutrients) of the mechanisms of the action of vitamin D on BCC, thus making the text more easily understandable to the Readers.

We have modified the formulation of the aim of the study.

We had made changes to the text of the Discussion to better align it with the changes in the Introduction.

The 3rd comment: The methods are too single. The levels of SHH and DBP should also be detected by Western blot, real-time quantitative PCR, and immunofluorescent staining.

Response: We thank the reviewer for this consideration. An objective inspection of the literature [Bouillon, R.; Schuit, F.; Antonio, L.; Rastinejad, F. Vitamin D Binding Protein: A Historic Overview. Front Endocrinol (Lausanne). 2020, 10, 910.]  in response to these concerns would seem to suggest that an Elisa technique by R&D used monoclonal antibodies was widely used to measure serum DBP concentration. The results obtained with this method surprised many experts as this assay showed race- and DBP/GC-specific results when the assay using polyclonal antibodies did not find racial differences in serum DBP. Further extensive studies, using polyclonal and mass spectrometry based assays convincingly demonstrated that the R&D monoclonal DBP assay discriminates against DBP/GC 1f and that all results should be carefully interpreted. Apart from ELISA assay, a wide variety of other assay methods, such as rocket electrophoresis, turbidimetry, nephelometry, and radial immunodiffusion have been developed and used. According to the design of the present research, we have planned to investigate the intracellular occurrence and distribution of DBP and SHH in heterogeneous samples that represented different types of BCC. We completed the research and concluded on the results observed at the protein level. Even though both qualitative and quantitative nucleic acid detection is widely used in medicine and biology in various applications, these were left behind the scope of the present research. With significant advantages of RT-PCR, including its ability to measure DNA concentrations over a large range, its sensitivity, its ability to process multiple samples simultaneously, etc. it lacks cellular specificity.

Studies on the contribution of DBP to skin cancers are almost lacking. Several studies reported on the necessity of cell-specific responses to vitamin D [Mull, B.; Davis, R.; Munir, I.; Perez, M.C.; Simental, A.A.; Khan, S. Differential expression of Vitamin D binding protein in thyroid cancer health disparities. Oncotarget. 2021, 12, 596-607; Vandikas, M. S.; Landin-Wilhelmsen, K.; Gillstedt, M.; Osmancevic, A. Vitamin D-Binding Protein and the Free Hormone Hypothesis for Vitamin D in Bio-Naïve Patients with Psoriasis. International journal of molecular sciences, 2022, 23, 1302.]. From this point of view, given that the DBP tissue assessment applied to different primary and recurrent BCCs bears novelty features. In this study, we coupled the tissue DBP and SHH indices confirmed in BCC subjects, thus adding a piece of knowledge of a complex relationship between vitamin D and DBP in BCC carcinogenesis.

Recently, the antibodies against tissue DBP were developed, allowing  FFPE sections to be treated with them. Following these recognized protocols we found a robust, significant, and consistent response in different primary and recurrent BCCs, the results of which achived in the given study were further submitted to statistical data analysis. In doing so, we have chousen an IHC visualization system based on chromogenic detection instead of fluorescent detection that is suitable for bright-field optics since the former has greater sensitivity and longer lasting signal, an important factor when the samples are collected prospectively, and a follow-up period is warranted. The essence of both assays is equal, and both methods, similarly to other assessment methodologies, display some advantages and disadvantages.

We are ready to keep the suggestions announced in mind when planning and designing further studies. Our thoughts regarding the assay methods used in the given study are accounted for and described in light of the research limitations.

The 4th comment: The results were not described clearly, especially the figure. In figure 2, the common criteria characteristic of BCC was not indicated in the figure 2A-D by arrows or others. In figure 6 and 7, the date was not described in the figures. Similarly, the same problem accrued in figure 8-10.

Response: We agree with the Reviewer. According to the Reviewer’s request, we have added symbols and hope that this will facilitate acceptance of the content related to the recognition of BCC-characteristic dermoscopic examination criteria to the Reader (Figure 2).

The dendrogram (Figure 6) consists of stacked branches (clades) that break down into further smaller branches. At the lowest level, individual elements appear and then they are grouped according to attributes into clusters with fewer and fewer clusters on higher levels. The end of each clade (a leaf) is the data. We have added this essential information to the legend describing this Figure.

A correlogram (Figure 7) is a graph of a correlation matrix. We have extensively used it since it allows us to highlight the most correlated variables in a data table. In this plot, correlation coefficients are colored according to the value. Positive correlations are displayed in blue, whereas negative correlations are in red. Color intensity is proportional to the correlation coefficients.

We have added this essential information to the legend describing this Figure.

Finally, Figures 8-10 are supplemented with essential information both textual and symbols, which does not repeat/overlap with the content included in the body of the text.

The 5th comment: The Discussion was mainly introduced previous study, did not see your discussion combined with your results.  According to your discussion, low level of serum vitamin D had been demonstrated in Latvia in 2015 by specialists from the Latvian Osteoporosis and Bone Metabolic Diseases Association, Riga Stradins University, and Riga East Clinical Hospital [38]. So we did not see your novelty, you should discuss your new findings in the discussion. Similarly, you detected the levels of SHH and DBP. As you known, vitamin D inhibits the SHH signaling pathway and displays a protective role in arresting tumor growth [37].  According to the previous report, the level of SHH increased when vitamin D decreased. Where is your new findings.   

Response: The authors appreciate the Reviewer's comment. As suggested, we have redrafted and rewritten the Discussion. To better highlight the novelty of the given paper the authors are ready to provide the following argumentative essays:

There have not been any vitamin D studies conducted in Latvia related to the oncology field. The merits of this study include the conclusive finding that 94.9% of the Latvian cohort of BCC patients were found to be deficient in vitamin D.The only study that has been done before in Latvia was performed in 2015 by specialists from the Latvian Osteoporosis and Bone Metabolic Diseases Association with the aim of the improvement of the treatment of osteoporosis and psoriasis.

The second issue to mention is the assessment of SHH tissue indices in conjunction with DBP indices and the levels of vitamin D observed in the same BCC subjects. Furthermore, the use of complex statistical evaluation, including the application of hierarchical clustering, allowed the recognition of the heterogeneity of the study cohort depicted as the presence of three major branches of the cluster tree in our BCC patient cohort.

Undoubtedly, the high practical value gained the conclusion that susceptible male individuals with low blood vitamin D levels were recognized at risk of developing aggressive and recurrent BCC and confirmed by the use of hierarchical clustering analysis and correlation analysis.

The 6th comment: There are some spelling mistakes, such as H2O2 in line 196 and 4 ◦C in line 199.

Response: We apologize to the Reviewer for the unfortunate spelling mistake. The typing errors were corrected.

The 7th comment: The description of results should be in sequence, figure 9 should be ahead figure 8, because you described figure 9 first in line 346. Similarly, in figure 8 you described figure 8D first in line 350.

Response: We thank the reviewer for his/her attentiveness. We made corrections in the body of the text and put Figures 8-10 straight-forward from 8 to 10 and using the appropriate sequence.

Reviewer 2 Report

In this paper, the authors presented the results of the study focused on determination of the serum levels of vitamin D in patients with primary and recurrent BCC of the head and neck and link these data to the results of tissue SHH and VDBP immunohistochemistry assessment. Undoubtedly, the study concerns a very important issue. Vitamin D level, mechanism of action and its role in cancerogenesis still attract attention of many researches. Design and conduction of presented study is proper.

During the reading of this article, I had minor suggestions and questions:

1.      How common are BCCs in Latvians?

2.      Are there any data supporting the use of vitamin D as an additive in the treatment of BCCs?

3.      Have the authors considered performing a regression analysis?

Author Response

The authors thank the reviewer very much for his time and expertise in the field, as well as for valuable suggestions related to the improvement of the manuscript. To the best of our knowledge and possibilities, we have followed the reviewer’s suggestions.

The 1st comment: How common are BCCs in Latvians?

Response: We thank the reviewer for such a valuable interest and comment, however, the authors should state that only limited information is accessible from the Latvian Cancer Register. In Latvia, unfortunately, BCC is not distinguised separately; it belongs to the group with the C44 diagnosis code, which includes all non-melanocytic skin tumors. The official reports from the Latvian Disease Prevention and Control Center (SPKC) apper with a certain delay. Currently, according to the Latvian Health Statistics Database, the information on the year 2017 is available: (https://statistika.spkc.gov.lv/pxweb/en/Health/Health__Saslimstiba_Slimibu_Izplatiba__Onkologija/ONKO040_2.px/table/tableViewLayout2/).However, the accumulating evidence suggests that the BCC incidence is constantly growing.

The 2nd comment: Are there any data supporting the use of vitamin D as an additive in the treatment of BCCs?

Response: The authors appreciate the Reviewer's interest and comment. There have not been any vitamin D studies conducted in Latvia related to the oncology field. The only study that has been done before in Latvia was performed in 2015 by specialists from the Latvian Osteoporosis and Bone Metabolic Diseases Association with the aim of the improvement of the treatment of osteoporosis and psoriasis. The results of the aforementioned study and the drawn conclusions have highlighted the presence of vitamin D deficiency in 82% of the study participants. The merits of the present study include the conclusive finding that 94.9% of the Latvian cohort of BCC patients were found to be deficient in vitamin D.

An objective inspection of the literature provides some knowledge about treatment options with vitamin D applied topically and vitamin D supplements in patients with skin cancers.

Furthermore, there is little information available about the use of vitamin D for the treatment of BCC when applied as a cream or a supplement, however, a combination of vitamin D and 5-FU has already been effectively used to treat actinic keratosis and squamous cell carcinoma. There is also evidence that an additional intake of vitamin D lowers the frequency of relapses. The list of publications available to the authors and related to the aforementioned issue appears below:

Ince, B., Yildirim, M.E.C., Dadaci, M. Assessing the Effect of Vitamin D Replacement on Basal Cell Carcinoma Occurrence and Recurrence Rates in Patients with Vitamin D Deficiency. HORM CANC, 2019,10, 145–149. https://doi.org/10.1007/s12672-019-00365-2

Rosenberg, A.R.,  Tabacchi, M.,  Ngo K.H.,  Wallendorf, M., Rosman, I.S., Cornelius, L.A.,  Demehri, S.. Skin cancer precursor immunotherapy for squamous cell carcinoma prevention. JCI Insight, 2019, 4:e125476. https://doi.org/10.1172/jci.insight.125476.

Kim, J. S., Jung, M., Yoo, J., Choi, E. H., Park, B. C., Kim, M. H., Hong, S. P. Protective Effect of Topical Vitamin D3 against Photocarcinogenesis in a Murine Model. Annals of dermatology2016, 28, 304–313. https://doi.org/10.5021/ad.2016.28.3.304

Azin, M., Mahon, A. B., Isaacman, S., Seaman, J. E., Allen, I. E., Szarek, M., Cornelius, L. A., Demehri, S. Topical Calcipotriol Plus 5-Fluorouracil Immunotherapy for Actinic Keratosis Treatment. JID innovations: skin science from molecules to population health2022, 2, 100104. https://doi.org/10.1016/j.xjidi.2022.100104

Seckin, D., Cerman, A. A., Yildiz, A., Ergun, T. Can topical calcipotriol be a treatment alternative in actinic keratoses? A preliminary report. Journal of drugs in dermatology: JDD2009, 8, 451–454. https://pubmed.ncbi.nlm.nih.gov/19537367/

The 3rd comment: Have the authors considered performing a regression analysis?

Response:  The authors thank the Reviewer for the comment. Regression analysis is a set of statistical processes for evaluating the relationship between a dependent variable and an independent variable. In our study, there were no comparable groups in which we could confidently single out one as a dependent group, and the other as an independent group, since basically all variables belonged to the latter. Therefore, to assess the relationship between the relevant variables, we conducted a correlation analysis, and the linear regression method was used just to better visualize the direction of the correlation.
